# Use of High Resolution Spatiotemporal Gene Expression Data to Uncover Novel Tissue-Specific Promoters in Tomato

Lulu Chen [1,2], Yuhang Li [1], Yuting Wang [1], Wenzhen Li [1], Xuechao Feng [1] and Lingxia Zhao [1,*]

1   Plant Biotechnology Research Center, School of Agriculture and Biology, Shanghai Jiao Tong University, Shanghai 200240, China; chenll@yctu.edu.cn (L.C.); l-sunshine@sjtu.edu.cn (Y.L.); yutingwang3103@gmail.com (Y.W.); liwz2022@hotmail.com (W.L.); mycjcaq@gmail.com (X.F.)
2   Jiangsu Key Laboratory for Bioresources of Saline Soils, Jiangsu Synthetic Innovation Center for Coastal Bio-Agriculture, School of Wetlands, Yancheng Teachers University, Yancheng 224007, China
*   Correspondence: lxzhao@sjtu.edu.cn; Tel.: +86-21-3420-5775

**Abstract:** Genetic modification can be an effective strategy for improving the agronomic traits of tomato (*Solanum lycopersicum*) to meet demands for yield, quality, functional components, and stress tolerance. However, limited numbers of available tissue-specific promoters represent a bottleneck for the design and production of transgenic plants. In the current study, a total of 25 unigenes were collected from an RNA-sequence dataset based on their annotation as being exclusively expressed in five type of tissues of tomato pericarp (outer and inner epidermis, collenchyma, parenchyma, and vascular tissues), and every five unigenes, was respectively selected from each tissue based on transcription expression. The 3-kb 5′ upstream region of each unigene was identified from the tomato genome sequence (SL2.50) using annotated unigene sequences, and the promoter sequences were further analyzed. The results showed an enrichment in T/A (T/A > 70%) in the promoter regions. A total of 15 putative tissue-/organ-specific promoters were identified and analyzed by real-time (RT) quantitative (q) PCR analysis, of which six demonstrated stronger activity than widely used tissue-specific tomato promoters. These results demonstrate how high spatiotemporal and high throughput gene expression data can provide a powerful means of identifying spatially targeted promoters in plants.

**Keywords:** *cis*-regulatory element; genetic improvement; omics; *Solanum lycopersicum*; tissue-specific promoter





## 1. Introduction

Tomato (*Solanum lycopersicum*) is at most a dual-purpose crop for vegetable and fruit worldwide, and it has also served as a model plant in many aspects of fruit physiology, development, and metabolism, as well as genetics, evolutionary biology, and developmental biology [1]. Currently, the progress of tomato breeding was prompted by molecular maker-assisted breeding techniques with molecular biology developments, and efficiency and the predicted breeding was sharply rapid with the biotechnological development and dissection of several metabolic pathways. It especially concerned the medical industry when tomato was used as a model system to express therapeutic proteins [2–4]. However, the content of the foreign protein was limited to a low-activity promoter that controls gene expression encoding. This limits tomato application as an expression system for the development of molecular farming.

A gene promoter usually, but not exclusively, comprises a DNA sequence upstream of the transcribed portion of the gene, and has sequences/sites that mediate the recognition and binding of the RNA polymerase, leading to a transcriptional initiation. Indeed, promoters with multiple *cis*-regulatory elements play critical roles in gene transcriptional expression regulation [5]. Thereafter, a large number of promoters have been identified by

isolating the target genes in a wide range of prokaryotes and eukaryotes and by analyzing the upstream sequences, and the characteristics and functions of the promoters have since been extensively investigated in model organisms [6,7]. Promoters have been widely deployed through genetic modification to alter organismal traits, such as crop yield and quality, and for the production of functional secondary metabolites that can be used as pharmaceuticals to benefit human health and to treat diseases [3,8–12].

Promoters can generally be classified into three categories: constitutive [13,14], tissue-specific [15,16], and inducible [17–19]. The constitutive promoters 35S (cauliflower mosaic virus 35S RNA) and Ubiquitin have been widely used to drive the expression of the genes of interest in dicotyledonous and monocotyledonous species, respectively [20–24]. However, the use of constitutive promoters can have negative effects, such as gene silencing, an excessive consumption of plant energy, disease symptoms, and effects on morphogenesis, growth, and development [25,26]. These factors limit the application of constitutive promoters when improving traits such as plant yield and quality [27–29]. As a result, researchers have come to recognize the potential advantages of organ- or tissue-specific promoters, such as avoiding the excessive accumulation of heterologous proteins in non-targeted tissues and the precise control of the expression of target genes according to predictable timing, localization, and even level of expression [30]. Accordingly, tissue-specific promoters have been used for improving agricultural traits and to drive the production of proteins and secondary metabolites in target organs/tissues in a process that has been referred to as molecular farming [4,29,31].

An example of specific promoters that have been characterized to date include the fruit-specific promoters from tomato (*Solanum lycopersicum*), such as those of the genes *E4*, *E8*, *PG*, and *2A11* [32–37], and the anther- and/or pollen-specific *LAT52* and *LAT59* gene promoters [15,38–40]. In other species, notable organ- or tissue-specific gene promoters include *RTS* from *Oryza sativa* [41], *pchiA*, and *pchiB* from *Petunia hybrida* [42], the seed-specific *Les4* promoter from *Vicia faba* [43], the stem-specific *SHDIR16* and *pSHOMT* promoters from a *Saccharum* hybrid [44], the *ScLSG* promoter from *Saccharum officinarum* [45], and the vascular tissue-specific *LlCCR* and *LlCAD* promoters from *Leucaena leucocephala* [46]. However, the number of robust tissue-specific promoters is limited and has not met the demand for plant molecular farming and the genetic improvement of specific crop traits [2,4].

However, Cortés et al. (2020, 2021) proposed that the genetics of adaptation to new environments is an important field for tolerance to abiotic stress in plant-genetics improvement [47,48]. This implies that the spatiotemporal gene expression maybe prefers epigenetic, rather than plastic. Manning et al. (2006) firstly found that a natural epigenetic mutation in a gene-encoding and SBP-box transcription factor alters tomato fruit color [49].

In this current study, to identify novel and highly efficient tissue-specific promoters, a total of 25 unigenes that were exclusively expressed in one of the five tissues (the top five from each tissue) were selected from a publicly available high spatiotemporal resolution RNA-seq dataset that was derived from five pericarp tissues of tomato fruit (*S. lycopersicum*, cv. Ailsa Craig) at 10 days post anthesis (dpa) [50], and 15 tissue-/organ-specific promoters were identified by an analysis of the expressional patterns of all 25 unigenes via real-time quantitative PCR (RT-qPCR). This provides a batch of highly efficient promoters to improve the agronomic traits of crops, and also exploit a kind of strategy to find novel, highly efficient tissue-/organ-specific promoters.

## 2. Materials and Methods

### 2.1. Promotor Sequence of Tissue-Specific Unigenes

We took advantage of a publicly available high spatiotemporal resolution RNA-seq dataset that was derived from five pericarp tissues: the oep (outer epidermis), col (collenchyma), par (parenchyma), vas (vascular tissues), and iep (inner epidermis) of tomato fruit (*S. lycopersicum*, cv. Ailsa Craig) at 10 dpa (Supplementary dataset S1) [50]. A total of 25 unigenes were chosen from among 20,976 high-quality expressed unigenes. Unigenes were identified that corresponded to the five most-abundantly expressed tissue-

specific genes from each of the five tomato fruit (cv. Ailsa Craig) pericarp tissues: oep, col, par, vas, and iep. Then, the 25 unigenes were mapped to the tomato genome (SL2.50; http://solgenomics.net/ accessed on 15 October 2021) to capture the 3 kb region sequence upstream of the initiation codon (ATG).

### 2.2. Promoter Sequence Analysis and Examination of cis-Regulatory Elements

The DNA sequence of each of the 25 unigenes was used to search the Sol Genomics Network database (http://solgenomics.net/ accessed on 15 October 2021) to identify the 3 kb upstream sequence from the ATG initiation codon. The genetic distance between the 25 promoter sequences was determined using the Molecular Evolutionary Genetics Analysis, v. 6.0 software package (Supplementary file S2). A cluster analysis was performed using the genetic distance values, and a heatmap was drawn using Multi-Experiment Viewer software [51]. The 3 kb promoter sequence from each unigene was submitted to the New PLACE database to identify regulatory elements (https://sogo.dna.affrc.go.jp/cgi-bin/sogo.cgi?lang=en&pj=640&action=page&page=newplace accessed on 15 October 2021) [52].

### 2.3. Plant Materials

Tomato (cv. M82) seeds, provided by the Tomato Genetics Resource Center (University of California, Davis, CA, USA, http://tgrc.ucdavis.edu/ accessed on 15 October 2021), were sown in 60-cell breeding plug trays (Taizhou Sophia Import & Export Co., Ltd., Taizhou, China) with humid peat pellets. Seeds were germinated at 26/20 °C (day/night) in a light-seeding box (SG650, Shanghai, China) and seedlings were grown under standard greenhouse conditions at the Pujiang Experimental Farm at the School of Agriculture and Biology, Shanghai Jiao Tong University (Shanghai, China). Seedlings were transferred to natural light polycarbonate greenhouse conditions when they had four true leaves.

### 2.4. RNA Extraction and Identification of Tissue-Specific Genes

Total RNA was extracted from tomato (cv. M82) pericarp at 10 dpa, 35 dpa (mature green stage), 47 dpa (breaker stage), and 54 dpa (red ripe stage) using the RNA prep pure Plant Kit (Tiangen, Beijing, China), according to the manufacturer's instructions. RNA was similarly extracted from different organs: roots, stems, leaves, and flowers. Each biological replicate comprised three individual plants, and a total of three biological replicates were collected. The concentration and quality of the total RNA samples were evaluated using a NanoDrop 2000 (Thermo Scientific, Waltham, MA, USA), and a total RNA of 1 μg was used as the template to synthesize first-strand cDNA with the PrimeScript™ RT Master Mix Kit (TaKaRa, Dalian, China). The cDNA was diluted 100-fold with RNase-free water to perform RT-qPCR with gene specific primers (Supplementary file S9) and the SYBR Premix Ex *Taq*™ II Kit (TaKaRa, Dalian, China) according to the manufacturer's protocol. The RT-qPCR program was 95 °C for 120 s, followed by 40 cycles of 95 °C for 20 s, 60 °C for 20 s, 72 °C for 20 s, and finally, 72 °C for 5 min, followed by a 10 °C hold, and the reactions were run on a Roche Light-Cycler® 96 (Roche, Germany). The *ACTIN* (GenBank: BT013524) gene was used as the reference for data normalization. The $2^{-\Delta CT}$ values were calculated, and statistically significant differences in the expression levels were determined using a Duncan's multiple range test [53].

## 3. Results

### 3.1. Identification of the Five Most-Abundantly Expressed Tissue-Specific Unigenes from Each of the Five Tomato Pericarp Tissues

A total of 624 tomato pericarp tissue-specific unigenes were reported by Matas et al. (2011), which were distributed as follows: oep (217), col (39), par (24), vas (284), and iep (60) (Figure 1 and Supplementary file S1). The authors also found that the tissue-specific unigenes were mostly distributed in both tissues of the oep and vas (501/624). We selected

the five most-abundantly expressed tissue-specific unigenes from each of the five tissues, giving a total of 25 genes (Table 1).

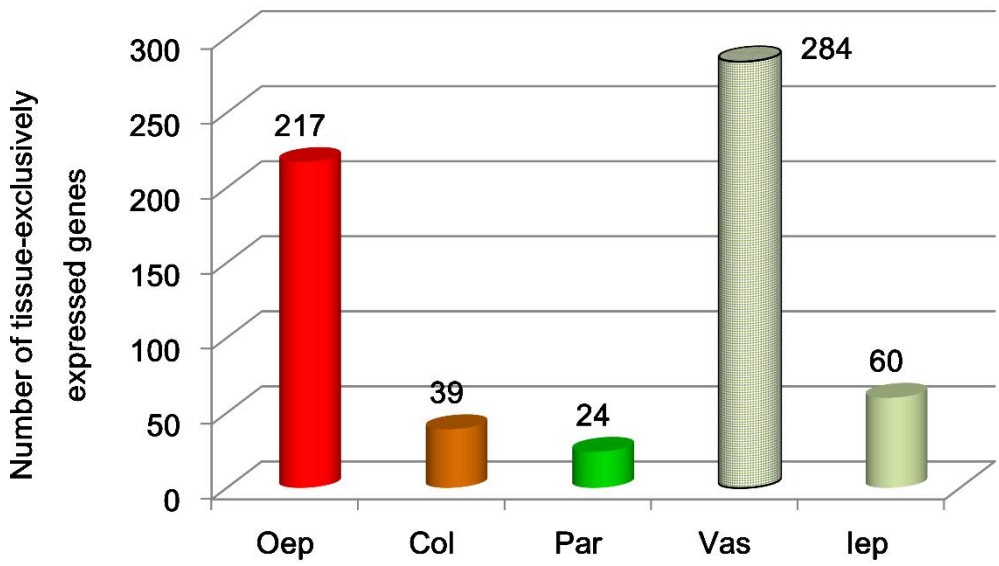

**Figure 1.** Number of unigenes exclusively expressed in one of the five tissues of tomato pericarp (cv. Ailsa Craig). oep, outer epidermis; col, collenchyma; par, parenchyma; vas, vascular; iep, inner epidermis. Data from Supplementary dataset 1 of Matas et al. (2011).

### 3.2. Promoter A/T Content and Cluster Analysis

The 3-kb promoter sequences upstream of the predicted ATG initiation codon of each of the 25 unigenes were identified in the Sol Genomics Network database (http://solgenomics.net/ accessed on 15 October 2021) (Supplementary file S2). The A/T content of each ~3000 bp region was calculated, as well as that of a series of promoter fragments (~2500 bp, ~2000 bp, ~1500 bp, ~1000 bp, and ~500 bp) (Supplementary file S3). The A/T content ranged from 42% (~2500 bp of *pSolyc06g024380.1*) to 78% (~1000 bp at *pSolyc01g067730.2*), and was >70% and higher than that for C/G, except for *pSolyc06g024380.1* in oep (Supplementary file S4). The average A/T content varied from 68.67% (~3000 bp) to 70.74% (~1000 bp) among the 25 promoters. The A/T content gradually increased, moving from ~3000 bp upstream of the ATG toward the ATG but decreased from ~1000 bp upstream of the ATG (Figure 2). The A/T-enriched promoter regions might facilitate double-strand melting for transcription initiation [54].

**Table 1.** The 25 unigenes expressed exclusively in one of the five tomato pericarp tissues (5 per tissue) *.

| Gene ID | Annotation | Locus | Promoter Region (−3000 to −1) |
|---------|------------|-------|-------------------------------|
| **Outer epidermis** | | | |
| *Solyc03g098700.1* | putative Kunitz-type tuber invertase inhibitor precursor (*Solanum tuberosum*) | SL2.50ch03:60981830..60981171 | SL2.50ch03:60984830..60981831 |
| *Solyc06g024380.1* | PREDICTED: uncharacterized protein LOC101232191 (*Cucumis sativus*) | SL3.0ch06:11249034..11249552 | SL3.0ch06:11246034..11249033 |
| *Solyc10g075090.1* | PREDICTED: non-specific lipid-transfer protein 1-like (*Solanum lycopersicum*) | SL2.50ch10:58801024..58800507 | SL2.50ch10:58804024..58801025 |
| *Solyc10g075100.1* | non-specific lipid transfer protein precursor (*S. lycopersicum*) | SL2.50ch10:58811212..58810582 | SL2.50ch10:8814212..58811213 |
| *Solyc10g076200.1* | non-specific lipid-transfer protein 2-like (*S. lycopersicum*) | SL2.50ch10:59051770..59052216 | SL2.50ch10:59048770..59051769 |
| **Collenchyma** | | | |
| *Solyc01g105040.2* | PREDICTED: uncharacterized protein LOC101251628 (*S. lycopersicum*) | SL2.50ch01:93331608..93329788 | SL2.50ch01:93334592..93331593 |
| *Solyc03g005000.2* | PREDICTED: protease HtpX homolog 2-like isoform 1 (*S. lycopersicum*) | SL2.50ch03:16712..19799 | SL2.50ch03:13817..16816 |
| *Solyc07g020860.2* | thioredoxin peroxidase 1 (*S. lycopersicum*) | SL2.50ch07:14295272..14300912 | SL2.50ch07:14292470..14295469 |
| *Solyc09g013150.2* | PREDICTED: probable anion transporter 3, chloroplastic-like (*S. lycopersicum*) | SL2.50ch09:5558428..5552677 | SL2.50ch09:5561282..5558283 |
| *Solyc10g006790.2* | PREDICTED: probable serine/threonine-protein kinase abkC-like (*S. lycopersicum*) | SL2.50ch10:1231572..1237111 | SL2.50ch10:1228572..1231571 |
| **Parenchyma** | | | |
| *Solyc01g067730.2* | PREDICTED: acyl carrier protein 1, chloroplastic-like (*S. lycopersicum*) | SL2.50ch01:76674532..76676885 | SL2.50ch01:76671781..76674780 |
| *Solyc01g096270.2* | PREDICTED: cytochrome b5-like (*S. lycopersicum*) | SL2.50ch01:87349969..87350651 | SL2.50ch01:87347174..87350173 |
| *Solyc04g007770.2* | PREDICTED: major latex-like protein (*S. lycopersicum*) | SL2.50ch04:1458908..1456588 | SL2.50ch04: 1461693..1458694 |
| *Solyc07g049140.2* | PREDICTED: metallocarboxypeptidase inhibitor, fruit-specific protein (*S. lycopersicum*) | SL2.50ch07:59369096..59367495 | SL2.50ch07:59371828..59368829 |
| *Solyc09g007940.2* | PREDICTED: adenosine kinase 2-like (*S. lycopersicum*) | SL2.50ch09:1440058..1444221 | SL2.50ch09:1437058..1440057 |
| **Vascular** | | | |
| *Solyc01g111310.2* | LAX2 protein (*S. lycopersicum*) | SL2.50ch01:97595955..97592414 | SL2.50ch01:97598905..97595906 |
| *Solyc04g026020.2* | PREDICTED: sieve element-occluding protein 3 (*S. lycopersicum*) | SL2.50ch04:19817933..19814414 | SL2.50ch04:19820933..19817934 |
| *Solyc05g006830.2* | PREDICTED: thioredoxin H2-like (*S. lycopersicum*) | SL2.50ch05:1454232..1453126 | SL2.50ch05:1457094..1454095 |
| *Solyc06g075220.1* | PREDICTED: fasciclin-like arabinogalactan protein 11-like (*S. lycopersicum*) | SL2.50ch06:46677877..46678626 | SL2.50ch06:46674877..46677876 |
| *Solyc09g010080.2* | invertase 5, an extracellular invertase (*S. lycopersicum*) | SL2.50ch09:3475480..3479343 | SL2.50ch09:3472522..3475521 |
| **Inner epidermis** | | | |
| *Solyc04g072310.2* | PREDICTED: uncharacterized protein LOC101262106 (*S. lycopersicum*) | SL2.50ch04:59344118..59341881 | SL2.50ch04:59346918..59343919 |
| *Solyc04g082170.2* | PREDICTED: alcohol dehydrogenase-like 7-like (*S. lycopersicum*) | SL2.50ch04:65946520..65949299 | SL2.50ch04:65943520..65946519 |
| *Solyc06g065530.2* | PREDICTED: GDSL esterase/lipase At1g29670-like (*S. lycopersicum*) | SL2.50ch06:40913948..40910209 | SL2.50ch06:40916912..40913913 |
| *Solyc09g062960.2* | PREDICTED: uncharacterized protein LOC101264365 (*S. lycopersicum*) | SL2.50ch09:60864690..60864224 | SL2.50ch09:60867472..60864473 |
| *Solyc09g062970.1* | PREDICTED: uncharacterized protein LOC101264365 (*S. lycopersicum*) | SL2.50ch09:60877001..60877273 | SL2.50ch09:60874001-60877000 |

* Data in the table was obtained from an RNA-seq dataset [50]; the top five unigenes exclusively expressed in one of the five tomato (cv. Ailsa Craig) fruit pericarp tissues at 10 dpa (days post anthesis) were selected.

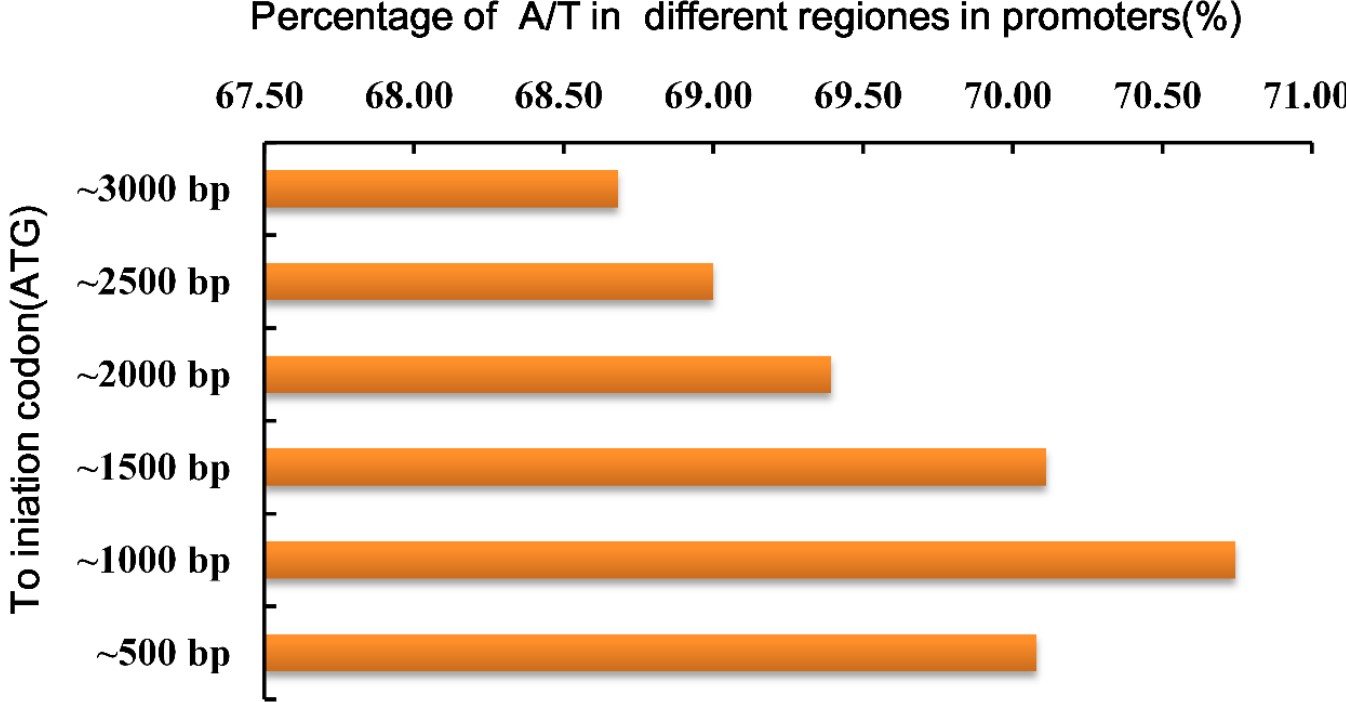

**Figure 2.** A/T content in different promoter regions, from ~3000 to ~1 bp. The data were collected from different promoter regions, ~3000 to ~1 bp from the 25 promoters; the 'A' of the translation initiation codon (ATG) was designated as bp + 1, while the next upstream 'A' nucleotide was designated bp −1.

Genetic distances were estimated between the promoter sequences and lengths (~3000 bp, ~2500 bp, ~2000 bp, ~1500 bp, ~1000 bp, and ~500 bp) using the pairwise distance function of the MEGA 6.0 software. The values ranged from 0.479 (*Solyc01g111310.2* and *Solyc07g020860.2*, ~500 bp) to 0.768 (*Solyc06g024380.1* and *Solyc09g010080.2*, ~1000 bp). However, average genetic distances varied from 0.659 (~1000 bp) to 0.668 (~3000 bp), and there did not appear to be distinct differences between promoter fragments (~500 bp to ~3000 bp) (Supplementary files S5 and S6). A cluster analysis revealed that the distal end (~1501−~3000 bp) of six of the promoters (*pSolyc06g024380.1*, *pSolyc09g007940.2*, *Solyc03g005000.2*, *pSolyc01g096270.2*, *pSolyc01g105040.2*, and *pSolyc04g082170.2*) was evolutionarily distant from the other 19 promoters, while the proximal end (~1500 bp) of *pSolyc06g024380.1* and *Solyc03g005000.2* was evolutionary distant from the other 23 promoters. The distal end of *pSolyc01g067730.2* and *pSolyc09g013150.2* was evolutionarily close to the other 23 promoters, while the proximal ends of *pSolyc01g067730.2* and *pSolyc07g020860.2* were close to the other 23 promoters (Figure 3 and Supplementary file S6). We concluded that there was a low level of homology between promoters that are specific to each individual tissue, or between promoters that are associated with different tissues.

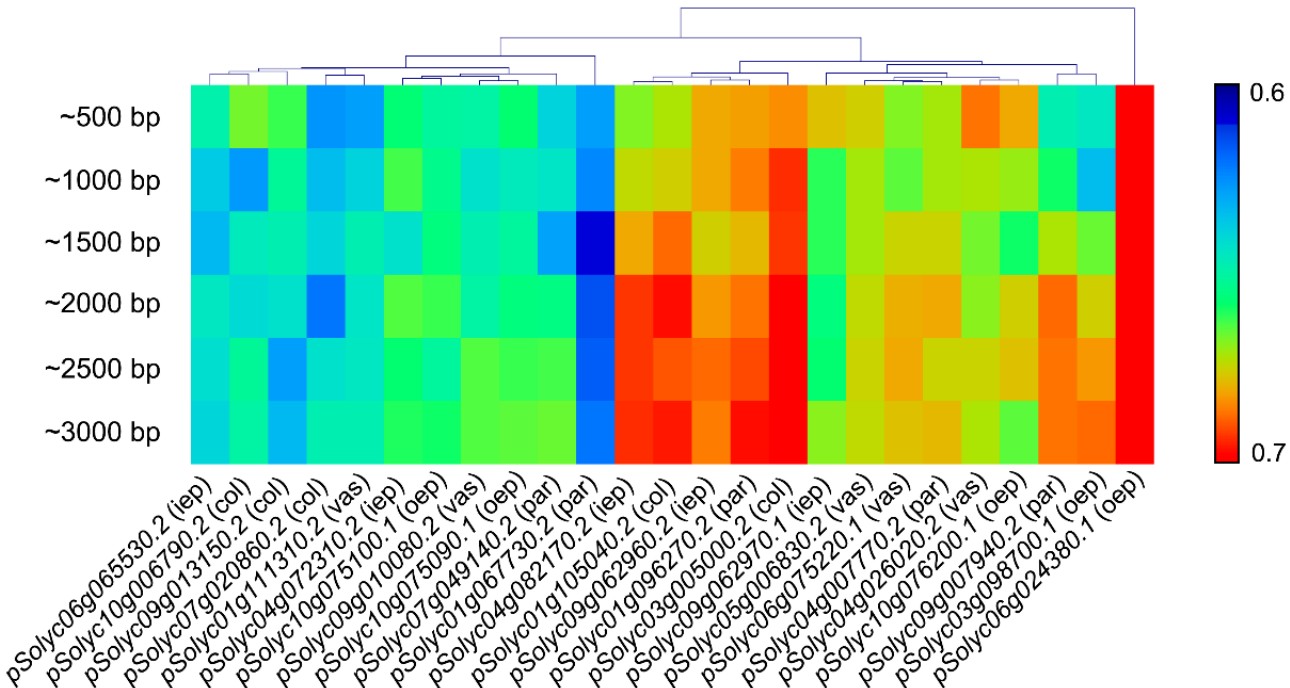

**Figure 3.** Cluster analysis heatmap drawn based on genetic distance values. The genetic distance between pairs of promoter sequences was calculated by the pairwise distance function of the MEGA 6.0 software package, and each genetic distance value in a heatmap line indicates the average from each promoter sequence compared to the other 24 (of the same size). Blue to red values on the scale represent genetic distances, from close to distant.

### 3.3. Expression Profiles of Pericarp Tissue-Specific Unigenes in Other Organs

The five most-abundantly expressed unigenes (*Solyc03g098700.1*, *Solyc06g024380.1*, *Solyc10g075090.1*, *Solyc10g075100.1,* and *Solyc10g076200.1*) associated with the oep had diverse expression profiles in different organs. The expression of *Solyc10g076200.1* was significantly higher in anthers (111.26) than in petals (58.70) and other organs (Figure 4a). *Solyc03g098700.1* was exclusively expressed in expanding green fruit (10 dpa), and both *Solyc10g075100.1* and *Solyc10g075090.1* were specifically expressed in leaves and expanding fruit (10 dpa). *Solyc06g024380.1* expression did not appear to be organ-specific and was expressed at higher levels in roots than in the other organs (Figure 4a).

Of the five unigenes (*Solyc01g067730.2*, *Solyc01g096270.2*, *Solyc04g007770.2*, *Solyc07g04-9140.2*, and *Solyc09g007940.2*) associated with the par tissue, both *Solyc04g007770.2* and *Solyc07g049140.2* showed fruit-specific expression, with *Solyc04g007770.2* exclusively expressed in the mature green stage (35 dpa), while *Solyc07g049140.2* was expressed during fruit development, reaching its peak at 54 dpa (red ripe stage). *Solyc09g007940.2* was expressed in anthers at significantly higher levels than in other organs. *Solyc01g096270.2* was expressed at high levels in flowers, and significantly higher in anthers than in other organs. *Solyc01g067730.2* was expressed in all organs investigated, and its expression in petals was significantly higher than in other organs (Figure 4b).

Two unigenes, *Solyc06g075220.1* and *Solyc04g026020.2*, selected from the vas tissue unigene set, were expressed in roots and stems at a significantly higher level than in other organs. *Solyc09g010080.2* was expressed in both anthers and carpels, and significantly higher than that in sepals, petals, and vegetative organs. Both *Solyc05g006830.2* and *Solyc01g111310.2* seemed to be constitutively expressed in all organs, but with significantly higher level in the leaves than in other organs (Figure 4c).

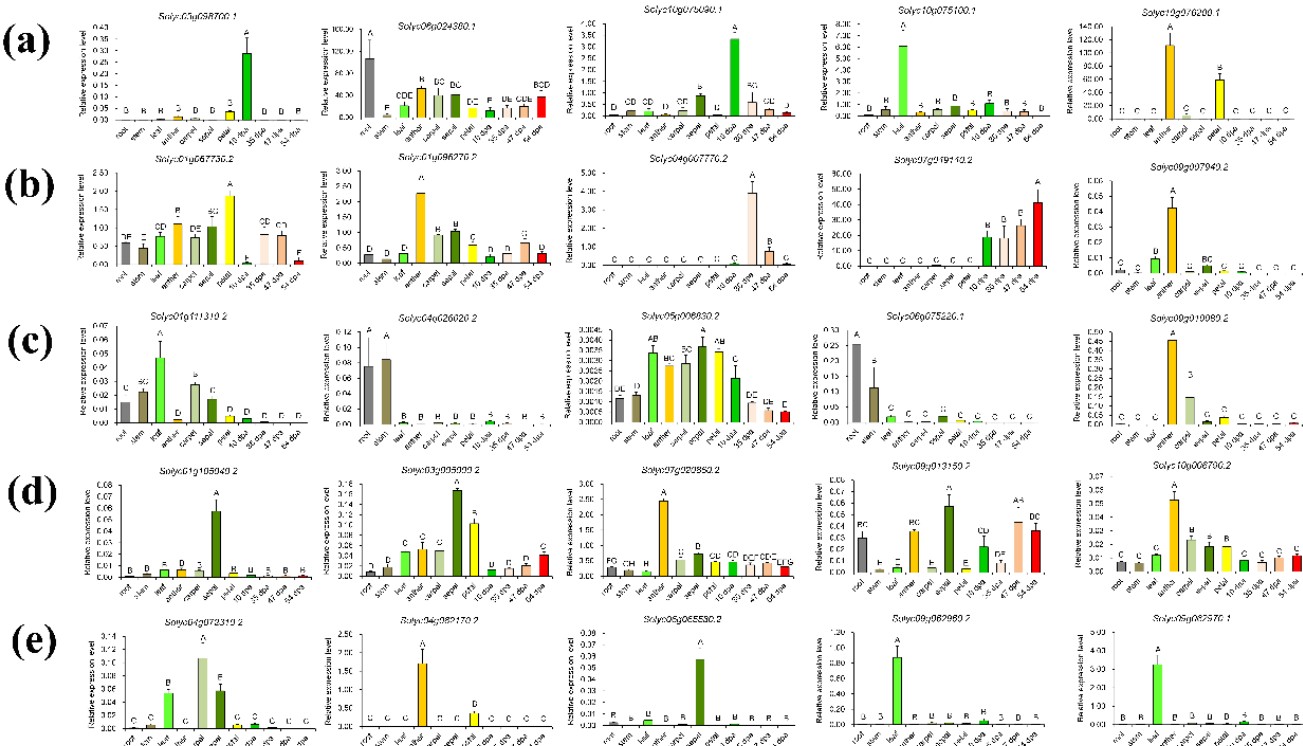

**Figure 4.** Transcriptional profiles of 25 tomato unigenes expressed in fruit, exclusively in one of the five pericarp tissues, across multiple organs. Expression profiles of 25 unigenes that were exclusively expressed in one of the five tissues (five from each tissue): outer epidermis (**a**), parenchyma (**b**), vascular (**c**), collenchyma (**d**), and inner epidermis (**e**). Real-time (RT) quantitative (q) PCR was used to examine expression in roots, stems, leaves, anthers, carpels, sepals, petals, and fruit (10 dpa, 35 dpa, 47 dpa and 54 dpa), with *ACTIN* as the reference gene. The data are the mean ± SD of three independent biological replicates (n = 3), and the capital letters indicate statistically significant differences between the measured values (*p* < 0.01), as determined by the Duncan's multiple range test, while lowercase letters indicate significant differences at the *p* < 0.05 level.

Of the five most-abundantly expressed genes associated with the col tissue (*Solyc07g02-0860.2*, *Solyc03g005000.2*, *Solyc10g006790.2*, *Solyc09g013150.2*, and *Solyc01g105040.2*), the expression of three (*Solyc01g105040.2*, *Solyc03g005000.2*, and *Solyc09g013150.2*) was significantly higher in the sepals than in other organs, with *Solyc01g105040.2* showing sepal-specific expression. Both *Solyc07g020860.2* and *Solyc10g006790.2* were expressed at a significantly higher level in anthers than in other organs, and *Solyc07g020860.2* showed anther-specific expression (Figure 4d).

Of the top five unigenes from the iep tissue (Solyc09g062970.1, Solyc09g062960.2, Solyc04g082170.2, Solyc04g072310.2, and Solyc06g065530.2), both Solyc09g062960.2 and Solyc09g062970.1 had leaf-specific expression. Solyc04g082170.2 and Solyc06g065530.2 were exclusively expressed in the anthers and sepals, respectively, and Solyc04g072310.2 was expressed in the carpels at statistically higher levels than in others organ, although expression was not carpel-specific (Figure 4e).

To summarize, a total of 15 tissue-specific genes were expressed exclusively or nearly exclusively in one of the investigated organs. The expression patterns were as follows: four in anthers (*pSolyc04g082170.2*, *pSolyc09g007940.2*, *pSolyc09g010080.2*, and *pSolyc10g076200.1*); four in fruit (*pSolyc03g098700.1*, *pSolyc04g007770.2*, *pSolyc07g049140.2*, and *pSolyc10g075090.1*); three in leaves (*pSolyc09g062960.2*, *pSolyc09g062970.1*, and *pSolyc10-g075100.1*); two in the roots/stems (*pSolyc04g026020.2* and *pSolyc06g075220.1*) and two in sepals (*pSolyc01g105040.2* and *pSolyc06g065530.2*) (Figure 4).

### 3.4. Distribution of cis-Regulatory Elements in Tissue-Specific Promoters

A total of 15 tissue-specific promoter sequences (~3 kb) were submitted to the New PLACE database to search for expressional regulatory elements. We focused on the three major promoter elements: core elements, elements associated with phytohormones, and those that confer tissue specificity (Figure 5, Supplementary files S7 and S8).

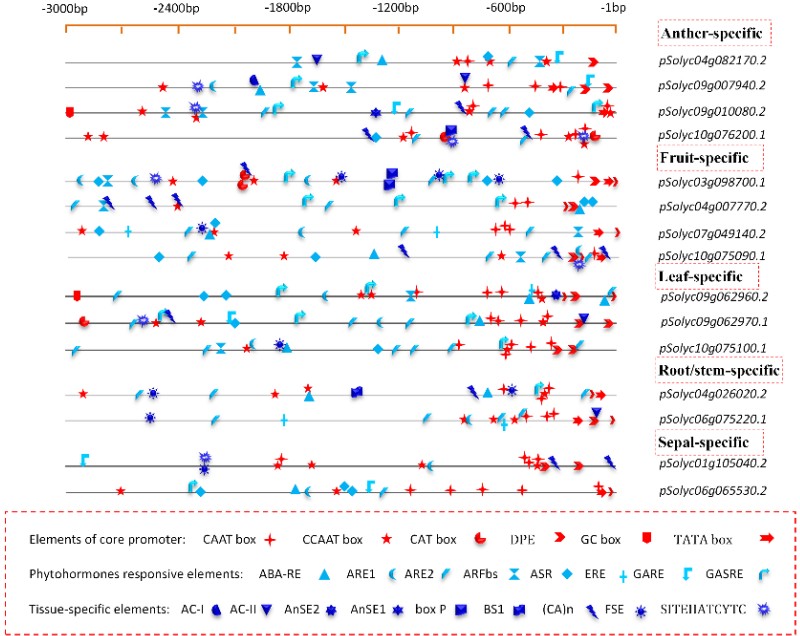

**Figure 5.** Distribution of *cis*-regulatory elements in different tissue-specific promoters. Elements in the core promoter: CAAT box (CAAT), transcriptional activation [55]; CCAAT box (CCAAT), transcriptional activation; CAT box (GCCAAC), *cis*-acting regulatory element related to meristem expression [56]; DPE, downstream promoter element (A/G$_{+28}$-G-A/T-C/T-G/A/C), located approximately 30 nucleotides downstream of the transcriptional start site (TSS) of many TATA-less promoters [57]; GC box (GGGCGG), enhancer [56]; TATA box (box1, CTATAAATAC; box2, TATAAAT; box4, TATATAA), transcription initiation [58]. Phytohormone-responsive elements: ABARE, abscisic acid (ABA)-responsive element (CATGCA) [59]; ARE1/ARE2, auxin-responsive element (CGTGG/ACTTTA) [59]; ARFbs, auxin response factor-binding site (TGt/cCTC), primary/early auxin response [60]; ASR, auxin- and salicylic acid-responsive site (TGACG) [59]; ERE, ethylene-responsive element (ATTTCAAA), senescence- and ethylene-responsive [61]; GARE (AAACAGA), gibberellin-responsive element [62]; GASRE, response to gibberellin, sugar element (TATCCA) [59]. Tissue-specific elements: AnSE1/AnSE2, anther-specific element (GAAT and ATTGTGA/TGTGGTT) [39,42]; AC-I/AC-II (ACCTACC/ACCAACC), responsible/enhanced xylem and/or repressed phloem for vascular tissue expression [63]; Box P (c/aACCAAAC), phenylpropanoid/lignin biosynthesis [64–66]; BS1 (AGCGGG), vascular tissue-specific expression [67]; (CA)n, (CNAACAC), embryo- and endosperm-specific transcription [68]; FSE (TGTc/tACA, fruit-specific expression, enhancer element) [69]; SITEIIATCYTC (TGGGCC/T), anther- and meristem-specific gene-expression element [70].

Core promoter elements were mostly located at the end proximal to ATG at ~−900 bp and included CAAT boxes (CAAT), downstream promoter elements (DPE, A/G$_{+28}$-G-A/T-C/T-G/A/C), GC boxes (GGGCGG), and TATA boxes (box1, CTATAAATAC; box2, TATAAAT; box4, TATATAA), while CCAAT boxes (CCAAT) were distributed along the entire promoter regions. The TATA box is an initiation site for transcription, and was found in eight organ-enriched promoters, included two anther-, three fruit-, two root-/stem-, and one sepal-specific promoter. Although the TATA box was not detected in the other seven

tissue-specific promoters, one to four copies of the DPE were found in these cases (Figure 5 and Supplementary file S7).

Plant hormone-responsive elements were present in all the promoters and were distributed at ~−2100 bp/~−3000 bp in the fruit-/leaf-specific promoters. Elements involved in auxin and gibberellin activity, such as ARE (auxin-responsive elements, ACTTTA or CGTGG), ARFbs (auxin response-factor binding site, TGt/cCTC) and GARE (gibberellin-responsive element motif, AAACAGA) were significantly more enriched than ABA-responsive elements (CATGCA) and EREs (ethylene-responsive elements, ATTTCAAA) (Figure 5 and Supplementary file S7).

Certain tissue-specific elements were also detected in some of the promoters, but none were found in *pSolyc06g065530.2*, a sepal-specific promoter (Figure 5 and Supplementary file S7). The anther-specific element SITEIIATCYTC (TGGGCY) was not only detected in the three anther-specific promoters (*pSolyc09g007940.2*, *pSolyc09g010080.2*, and *pSolyc10g076200.1*), but was also found in two fruit- (*pSolyc03g098700.1* and *pSolyc10g07509-0.1*), one leaf- (*pSolyc09g062970.1*), and one root-/stem- (*pSolyc01g105040.2*) associated promoter. An anther-specific element (GAAT/ATTGTGA), known from the *pchiB* (*CHALCONE FLAVANONE ISOMERASE B promoter*) gene [42], was detected in *pSolyc09g010080.2*, whereas an anther-specific pB core motif (TGTGGTT) [39] was found in the promoter of the leaf-specific *pSolyc09g062960.2*. The TGTc/tACA motif, a fruit-specific element, was detected in six tissue-specific promoters, which included one leaf- (*pSolyc10g075100.1*), two root-/stem- (*pSolyc04g026020.2* and *pSolyc06g075220.1*), one sepal- (*pSolyc01g105040.2*), and two fruit-specific (*Solyc03g098700.1* and *Solyc07g049140.2*) promoters.

The conserved C/AACCAAAC sequence from the Box P motif has been associated with lignin biosynthesis in monocotyledonous species such as rice (*Oryza sativa*), maize (*Zea mays*), and sorghum (*Sorghum bicolor*), and was also detected in the anther-specific promoter of *pSolyc10g076200.1* and the fruit-specific promoter of *pSolyc03g098700.1* in our tomato data. The conserved BS1 (Biding site 1) motif (AGCGGG) has been shown to be required for vascular-specific expression, and was only detected in a root-/stem-specific promoter (*Solyc04g026020.2*). The AC-I motif (ACCTACC), known to be responsible for driving expression in the xylem, was detected in the anther-specific *pSolyc09g007940.2* promoter, whereas AC-II (ACCAACC), which is known to be highly expressed in the xylem and to have low expression in the phloem, was found in two anther-specific (*Solyc09g007940.2* and *Solyc04g082170.2*), one leaf-specific (*Solyc09g062970.1*), and one root-/stem-specific promoter (*Solyc06g075220.1*) (Figure 5 and Supplementary file S7).

## 4. Discussion

The identification of tissue-/organ-specific promoters in plants would be beneficial for the genetic improvement of economically important traits, as well as the development of plant-derived functional products, such as antibodies, vaccines, and/or proteins for clinical use [3,4,8,9,31]. However, to date, the limited number and weak expression of known tissue-/organ-specific promoters limited their utilization in such plant-improvement programs. The exploitation of high resolution gene-expression data provides a potential path to the identification of novel promoters or the facilitation of the development of tissue-/organ-specific synthetics. Substantial amounts of gene-expression data are available for tomato, including transcriptome datasets [50,71]. Of the 25 tomato fruit-tissue-specific unigenes [50], we verified 15 tissue-/organ-specific promoters by RT-qPCR, which corresponds to 60% (15/25) of the identified genes (Table 1 and Figure 4). We also found that there was a low level of DNA sequence homology in the promoters by cluster analysis sequences (Figure 3).

Many functional elements have been identified in plant promoters [32,33,39,43,44,46,55, 67,72,73], and this has resulted in the development of the PLACE database (https://sogo.dna. affrc.go.jp/cgi-bin/sogo.cgi?lang=en&pj=640&action=page&page=newplace accessed on 15 October 2021) [52]. However, there is still an elementary understanding of which elements or conserved motif sequences precisely and effectively regulate the expression of the target

gene [41,74]. Moreover, strong, especially tissue-/organ-specific, promoters have not always been advantageous for improving all agricultural traits or in the production of therapeutic proteins in molecular farming [25,26].

Here, the activity of 15 tissue-/organ-specific promoters was compared to common tissue-specific promoters (*pLAT52*, *p2A11*, *pE8*, *pE4*, and *pPG*) [32,33,35–38,75,76] and a constitutive promoter (CaMV 35S) [20,21]. We found that an anther-specific promoter, *pSolyc10g076200.1*, had a 15-fold higher expression than the *LAT52* promoter [15], and a 35-fold higher expression than a *2×35S* promoter (Table 2) [14,20,21,77]. Three fruit-specific promoters (*pSolyc04g007770.2*, *pSolyc07g049140.2*, and *pSolyc10g075090.1*) showed 3- to 73-fold stronger expressions than the *E8* and *PG* promoters, and *pSolyc07g049140.2* had an 8-fold higher expression than the *2×35S* promoter (Table 2). *pSolyc10g075100.1* was expressed 1.91-fold stronger than the *2×35S* promoter, and *pSolyc09g062970.1* was as strong as the *2×35S* promoter (Table 2). The expression of a root-/stem-specific promoter, *pSolyc06g075220.1*, was 2.5-fold higher than that of the *pRSS1* promoter [78]. The enhancer element for fruit-specific expression, TGTc/tACA, was found in *pSolyc07g049140.2*, but not in *pSolyc10g075090.1* or *pSolyc04g007770.2* (Figure 5). This suggests that other fruit-specific elements are present in addition to TGTc/tACA. Pollen- and/or anther-specific motifs were also not detected in *pSolyc04g082170.2*, an anther-specific promoter (Figure 5 and Supplementary file S7). The vascular-specific AC-II (ACCAACC) element was only found in the promoter of the leaf-specific gene *pSolyc09g062970.1* (Figure 5 and Supplementary file S7). These results suggest that many of the tissue-/organ-specific elements have not yet been identified, and that complex regulatory systems underlie tissue- and/or organ-specific expression.

**Table 2.** Activity of tissue-specific promoters (*TSPs*) compared with commonly used promoters (*CPs*).

| *TSP* ID * | Ratio (*TSP/CP* **) | | | |
|---|---|---|---|---|
| | *pE8* | *pPG* | *pLAT52* | *2×35S* |
| **Anther-specific** | | | | |
| *pSolyc04g082170.2* | | | 0.23 | 0.54 |
| *pSolyc09g007940.2* | | | 0.01 | 0.01 |
| *pSolyc09g010080.2* | | | 0.06 | 0.14 |
| *pSolyc10g076200.1* | | | 15.32 | 35.07 |
| **Fruit-specific** | | | | |
| *pSolyc03g098700.1* | 0.03 | 0.20 | | 0.02 |
| *pSolyc04g007770.2* | 0.46 | 3.35 | | 0.38 |
| *pSolyc07g049140.2* | 10.05 | 72.98 | | 8.22 |
| *pSolyc10g075090.1* | 0.42 | 3.03 | | 0.34 |
| **Leaf-specific** | | | | |
| *pSolyc09g062960.2* | | | | 0.28 |
| *pSolyc09g062970.1* | | | | 1.02 |
| *pSolyc10g075100.1* | | | | 1.91 |
| **Root-/stem-specific** | | | | |
| *pSolyc04g026020.2* | | | | 0.02/0.03 |
| *pSolyc06g075220.1* | | | | 0.08/0.04 |
| **Sepal-specific** | | | | |
| *pSolyc01g105040.2* | | | | 0.05 |
| *pSolyc06g065530.2* | | | | 0.02 |

* In the column *TSP* ID, the *p* in front of a gene ID indicates that it is the promoter region of that target gene. ** Both *pE8* and *pPG* are tomato fruit-specific promoters. *pLAT 52* is an anther-specific promoter with a LAT 52 box; *2×35S* is a constitutive promoter with a double repeat of the 35S promoter sequence. The value for each promoter activity in the table has been normalized by the expression of the *ACTIN* gene. The expression values of *pE8*, *pPG,* and the 15 tissue-specific promoters were obtained by RT-qPCR. The activity of *pLAT52* was estimated from six flower RNA-seq data points (C34, C36, C37, C38, C39, and C40) on the TomExpress website [77] (Zouine et al. 2017). The activity of *2×35S* was estimated by overexpression of the *PHYTOENE SYNTHASE 1* (*PSY1*) gene in a yellow fruited cherry tomato and transcriptome data (unpublished data from our lab). *ACTIN* (GenBank accession BT013524) expression was always used as a reference gene.

Here, what we must emphasize is that 25 unigenes collected from an RNA-sequence dataset of the tomato cultivar Ailsa Craig [50], while the M82 was used to identify 15 tissue-/organ-specific promoters in the present research, the single nucleotide polymorphisms (SNP), or a few nucleotide differences of the promoter derived from the orthologs gene, may be residing between two tomato cultivars. This case was also observed at E8 promoter in tomato, although the identity of the E8 promoter DNA sequence reaches 99.64% among four cultivars of tomato (*Solanum lycopersicum*, Zaofen No.2, Zhongshu No.5, Red cherry, and Cherry) [29]. Therefore, we must concentrate on whether the mutations or different nucleotides fall to key *cis*-elements in the promoters.

Simultaneously, all of DNA methylation, histone modification and non-coding RNA regulation exhibit epigenetic aspects, and suppress the expression of target genes [79]. Although DNA methylation is tissue-specific, however, a great scientific problem is to clarify the relationship between DNA methylation regions and tissue-specific expressed genes [80]. In addition, DNA methylation also affects alternative splicing of mRNA at the post-transcriptional level [81].

To excavate tissue-specific promotors more effectively, an expression genome-wide association study (eGWAS) across several databases would be a more effective strategy than RNA-sequence, and LD patterns within the associated *cis*-regulatory promoter-linked regions of previous eGWAS, especially, would be an alternative approach to find active promoters in the future [82]. Of course, point mutation or single nucleotide polymorphisms frequently occur between orthologous genes in plants, even different cultivars. The slight difference in E8 promotors was previously observed in the cultivated cultivars of tomato (*S. lycopersicum*) [29]. This mutation would affect promoter activity, especially when it falls into *cis*-regulatory elements. Plasmids of the *RCc3:OsNAC5* and *GOS2:OsNAC5* were used to transform rice (*Oryza sativa*), and *RCc3:OsNAC5* plants showed a significantly higher grain yield of 22–63%, while the *GOS2:OsNAC5* plants showed a reduced or similar yield to the nontransgenic (NT) controls [83]. Yang et al. (2021) found that the relevant transcription factors bind to the *cis*-regulatory elements within the promoters of auxin transporter genes to respond to diverse stresses in transformed potato (*Solanum tuberosum*) [84]. However, a differential expression of the auxin transporter genes under abscisic acid and abiotic stresses indicated their specific adaptive mechanisms regulating tolerance to environmental stimuli. Wang et al. (2015) created the artificial green-tissue-specific promoters by assembling several DNA regulatory sequences, which include some *cis*-elements associated with tissue-specific expression [85]. Despite the novel promoters showing the property of the green-tissue-specific ones, the different expression efficiencies of GUS genes derived by those novel promotors occurred between various tissues. All of those results showed the relationship between specific stresses and potential shared genomic bases; this also especially exhibited the pleiotropic and epistatic effects that are associated with epigenetic Gene-Environment Interreaction (GxE) and plasticity effects mediated by the promoter.

## 5. Conclusions

Genetic modification can be an effective method for the improvement of plant traits, and in this context, the promoter region is an essential *cis*-regulatory element for the expression of target genes. Genome walking or thermal asymmetric interlaced polymerase chain reaction (TAIL-PCR) have, until recently, been the primary strategies for sequencing the promoters of genes of interest [56]. This approach is time consuming and laborious and the resulting limited set of tissue-/organ-specific promoters has not met the demands of biotechnological applications. However, this bottleneck may be addressed through the use of large datasets derived from analysis of genomes, transcriptomes, proteomes, and metabolome, which are being reported at an increasing rate, in line with technological developments.

We chose the model plant, tomato, an economically important crop, and designed an effective pipeline for identifying tissue-/organ-specific promoters. Based on a previously published RNA-seq dataset [50,71] created from five tomato fruit pericarp tissues (outer

and inner epidermal layers, collenchyma, parenchyma, and vascular tissues), a total of 25 genes were chosen from 20,976 high-quality unigenes, and mapped onto the tomato genome sequence (http://solgenomics.net accessed on 15 October 2021) to identify the promoter sequence for each. A total of 15 tissue-specific promoters were then verified as organ-specific by RT-qPCR, and six of these as more strongly expressed than other commonly used organ-specific promoters, such as fruit- and/or anther-specific promoters. This study lays a foundation for investigating the functions of target genes and dissecting the roles of *cis*-regulatory elements in different promoters.

**Supplementary Materials:** The following are available online at https://www.mdpi.com/article/10.3390/agriculture11121195/s1, Supplementary file S1: Number of tissue-specific genes; Supplementary file S2: Promoters (DNA sequence 3 kb upstream from the ATG initiation codon). A total of 25 promoters were obtained from five tissues (targeting five per tissue): outer epidermis (oep), collenchyma (col), parenchyma (par), vascular (vas), and inner epidermis (iep); Supplementary file S3: Number of A, T, C, and G bases in different regions of the 25 tissue-specific promoters; Supplementary file S4: Percentage (%) of A/T and C/G in different regions of the 25 tissue-specific promoters; Supplementary file S5: Average genetic distance between different lengths of the 25 promoters; Supplementary file S6: Genetic distance between each pair of promoter sequence;. Supplementary file S7: Distribution of *cis*-regulatory elements in the tissue-specific promoters. Supplementary file S8 Number of *cis*-regulatory elements in the tissue-specific promoters; Supplementary file S9: Primers used for RT-qPCR in this study.

**Author Contributions:** L.Z. conceived and designed the research and wrote the manuscript. L.C. performed the main experimental work, data analysis, and wrote the manuscript. Y.L. performed the RT-qPCR experiment on the tissues of the outer epidermis and inner epidermis. Y.W. collected and prepared the samples. Y.W. performed the RT-qPCR experiment on the tissues of parenchyma. W.L. analyzed the RT-qPCR data. X.F. participated in the genetic distance analysis. All authors have read and agreed to the published version of the manuscript.

**Funding:** This research was funded by the National Natural Science Foundation of China (31872112, 32072583), the Key Technology Research and the Development Program of Shanghai Technology Committee, Shanghai, China (19391904100), the SJTU Global Strategic Partnership Fund (2019 SJTU-HUJI, WF610561702), and the Jiangsu Provincial Double-Innovation Doctor Program, Jiangsu Province, China [JSSCBS20211141].

**Institutional Review Board Statement:** Not applicable.

**Informed Consent Statement:** Not applicable.

**Data Availability Statement:** Data is contained within the article and Supplementary Material.

**Acknowledgments:** The authors are grateful to Dani Zamir and the TGRC (Tomato Genetic Resource Center, UC Davis, CA, USA) for providing the tomato seeds. We are particularly grateful to Lida Zhang (School of Agriculture and Biology, Shanghai Jiao Tong University, Shanghai, China) for the cluster analysis. We are grateful to Jocelyn K. C. Rose (Plant Biology Section, School of Integrative Plant Science, Cornell University, Ithaca, NY, USA) for editing the manuscript and providing constructive suggestions for the manuscript.

**Conflicts of Interest:** The authors declare that they have no conflict of interest.

## Abbreviations

ARE: auxin responsive elements; ARFbs: auxin response factor-binding site; BR stage: breaker stage; BS1: Biding site 1; col: collenchyma; dpa: days post anthesis; DPE: downstream promoter element; ERE: ethylene-responsive element; GARE: gibberellin-responsive element motif; iep: inner epidermis; kb: kilo base; MG stage: mature green stage; oep: outer epidermis; par: parenchyma; pchiB: CHALCONE FLAVANONE ISOMERASE B promoter; RR stage: red ripe stage; RT-qPCR: real time quantitative PCR; TAIL-PCR: thermal asymmetric interlaced polymerase chain reaction; vas: vascular.

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
