# Peer review of "Use of High Resolution Spatiotemporal Gene Expression Data to Uncover Novel Tissue-Specific Promoters in Tomato"

_agriculture, doi:10.3390/agriculture11121195_

Round 1
Reviewer 1 Report
The article “Use of high resolution spatiotemporal gene expression data to uncover novel tissue-specific promoters in tomato” Lulu Chen, Yuhang Li, Yuting Wang, Wenzhen Li, Xuechao Feng, and Lingxia Zhao is devoted to the study of the spatial-temporal expression of a certain set of different genes of tomato, whose coding sequences were previously identified. The investigation was carried out by a combination of molecular techniques and bioinformatics methods. The work was done at a high level and leaves an excellent impression. The collective of authors has performed a tremendous research and received a large amount of valuable data. Undoubtedly, the identification of the expression patterns of selected genes is of great value for the practical application of the verified promoters of these genes in plant genetic engineering. And only for this reason, this study is worthy of publication in IJMS. The analysis of cis-motifs in the upstream promoter region of genes, carried out by the authors, is also a very valuable acquisition. For example, this will allow a more thorough study of the contribution of these cis-sequences to the expression of marker genes in future research.
Some minor notes.
Line 14-15. Please rephrase the sentence "their annotation as being exclusively expressed in fruit, in one of five pericarp tissues (outer and inner epidermis, collenchyma, parenchyma, vascular tissues, and 5 unigenes per tissue)" in the Abstract, as its meaning may not be clear enough for readers.
The research is mainly based on the work of Matas et al. 2011 in which the tomato cultivar Ailsa Craig was used. However, in present study, the authors used tomato cultivar M82. It is quite reliable that the coding sequences of the studied genes coincide in both cultivars. However, it may well be that the upstream promoter sequences of the same gene may differ slightly in different varieties of the same botanical species. Is it so? It is desirable to reflect this issue in the Discussion section at least in 2-3 sentences. And it is advisable to refer to some reference related to this issue.
Author Response
Dear Reviewer,
Thank you for your positive and constructive comments in relation to the manuscript entitled 'Use of High Resolution Spatiotemporal Gene Expression Data to Uncover Novel Tissue-Specific Promoters in Tomato'. It is appreciated.
Please see the attachment with detailed changes highlighted in green on manuscript addressing concerns raised.
Kind regards,
Authors

Reviewer 2 Report
The work by Chen et al. studied novel tissue-specific promoters in tomato using high resolution spatiotemporal gene expression [i.e. real time (RT)-quantitative (q)PCR analysis]. They found that six (out of 19) putative tissue/organ-specific promoters had stronger activity than widely used tissue-specific tomato promoters. Holistically, the manuscript is well framed and questions key scientific goals relevant for the plant molecular biology discipline.
A key pro of the study is its coherence. Yet, a more careful read invites authors to be more careful at conveying how well designed the sampling is (i.e. power analysis in terms of experimental units and replicates), while clearly drawing the line at the Introduction (last paragraph, where objective is stated) and the Discussion sections on what novelty this report brings. Besides, since spatiotemporal gene expression presumably exhibits a strong environmental interaction, authors must refer from the very begging to previous quantifications of this effect. A general recommendation is to first comment on this aspect before moving into a full RT-qPCR analysis, particularly for a plastic property such as spatiotemporal gene expression.
As a second, more technical, point, methods, results and data interpretation are generally coherent, and conclusions are well drafted. Still, some details are missing, such as the motivation to specifically choose the pool of putative tissue/organ-specific promoters. Authors must also embrace clearer considerations regarding linkage disequilibrium (LD) patterns (refer to and cite Tree Genet Genomes 2012 8(4):821-29, as a proxy for cis-regulation via promoters. Furthermore, I encourage authors to discuss the statistical power explicitly in the Discussion section by adding a paragraph on possible caveats, by performing an a priori power analysis. Some further suggestions are enlisted below.
- Please add to the introduction (as well as to the discussion section) brief paragraphs commenting on whether spatiotemporal gene expression can so far be understood in tomato as plastic, or rather epigenetic. For the latter, other alternatives beyond are already well studied, such as trans-generational epigenetic inheritance (although I recognize that until now it has played a role more as a ‘proof of concept). In order to merge these considerations with an integrated framework for molecular breeding of tomato, please also refer to recent trans-disciplinary developments for crop breeding reviewed in Front Plant Sci 2020 11:583323 (refer to and cite these works).
- I must confess that authors did not use more appealing approaches to optimize the pool of putative tissue/organ-specific promoters. Authors should explore (at least discuss within a new Perspectives section before Conclusions on L367) alternative strategies in which they prioritize putative tissue/organ-specific promoters. As a first step they could try a somehow innovative step in which preliminary ranking is carried out relying on previous eGWAS studies. Clearly, associated SNP markers from eGWAS studies may disregard SNPs with lower effects but that as a whole contribute accounting for the missing heritability (i.e. infinitesimal model, refer to and cite Tree Genet Genomes 2021 17:12). Therefore, better approaches to be implemented are: (1) weighted models using effect estimates gathered from previous studies, and (2) optimization by computing saturating curves of the predictive ability and LD patterns within the associated cis-regulatory promoter-linked regions of previous eGWAS.
- Not being my self an expert in tomato molecular breeding, I leave the corresponding details in the hands of more specialized reviewers, although I believe many details are still missing in the Introduction and Discussion sections, and therefore should be incorporated. A brief paragraph explaining the state of the art and perspective of tomato molecular breeding would help to contextualize readers coming more from the breeding side of other plant species.
- Last but not least, regarding pleiotropic and epistatic effects mediated by promoters, I encourage authors to discuss whether there could be genomic context-dependent effects and trade-offs between the studied tissue/organ-specific promoters with key pathways in plants. Or instance, please refer to and cite specific stresses with a potential shared genomic bases such as: drought , mineral deficit , and broad abiotic adaptation. These types of interactions, which in some cases could contribute to promoter-mediated epigenetic GxE and plasticity effects (via antagonistic pleiotropy or conditional neutrality), would better enlighten the overlap of the analyzed tissue/organ-specific promoters with overall plant responses.
Author Response

(The authors gave the same response as above.)

Reviewer 3 Report
The manuscript intends to fill a gap in the availability of promoters that can be used for transformation of tomato or other crops and delivers a good review of the topic, sound research design, and interesting findings. I am guessing the Authors are involved in tomato research and I would like a paragraph (Discussion) on how these 15 promoters/genes or the top 6 with high expression levels could be applied in the context of tomato. The Abstract could benefit also from having a sentence that specifies the tissues/organs that the promoters are associated with.
A few minor edits to care for as specified in the attached copy of the manuscript.

Author Response

(The authors gave the same response as above.)

Round 2
Reviewer 2 Report
Thanks for the thoughtful and precise amendments and description at the rebuttal letter. I do agree with the argumentation raised by the authors. My only comment left is that in the rebuttal letter the claimed that "Article (Front Plant Sci 2020 11:583323) had been cited in the new version MS". However, the reference list does not mention it. Therefore, please include the new citation (introduction would be a good place for that) so that the description within the rebuttal letter matches the new version of the manuscript.
Author Response
Dear Reviewer,
Thank you for your positive and constructive comments in relation to the manuscript entitled 'Use of High Resolution Spatiotemporal Gene Expression Data to Uncover Novel Tissue-Specific Promoters in Tomato'. It is appreciated.
Please see the attachment with detailed changes highlighted in yellow on manuscript addressing concerns raised.
Kind regards,
Authors
